# Fetal Bowel Abnormalities Suspected by Ultrasonography in Microvillus Inclusion Disease: Prevalence and Clinical Significance

**DOI:** 10.3390/jcm11154331

**Published:** 2022-07-26

**Authors:** Yue Sun, Changsen Leng, Sven C. D. van Ijzendoorn

**Affiliations:** 1Department of Biomedical Sciences of Cells and Systems, Section Molecular Cell Biology, University of Groningen, University Medical Center Groningen, 9713 AV Groningen, The Netherlands; y.sun@umcg.nl (Y.S.); lengchangsen@hotmail.com (C.L.); 2Center for Liver, Digestive & Metabolic Disease, University of Groningen, University Medical Center Groningen, 9700 AD Groningen, The Netherlands; 3State Key Laboratory of Oncology in South China, Collaborative Innovation Centre for Cancer Medicine, Guangdong Esophageal Cancer Institute, Department of Thoracic Surgery, Sun Yat-sen University Cancer Centre, Guangzhou 510060, China

**Keywords:** microvillus inclusion disease, prenatal bowel abnormalities, congenital diarrhea

## Abstract

Microvillus inclusion disease (MVID) is a rare, inherited, congenital, diarrheal disorder that is invariably fatal if left untreated. Within days after birth, MVID presents as a life-threatening emergency characterized by severe dehydration, metabolic acidosis, and weight loss. Diagnosis is cumbersome and can take a long time. Whether MVID could be diagnosed before birth is not known. Anecdotal reports of MVID-associated fetal bowel abnormalities suspected by ultrasonography (that is, dilated bowel loops and polyhydramnios) have been published. These are believed to be rare, but their prevalence in MVID has not been investigated. Here, we have performed a comprehensive retrospective study of 117 published MVID cases spanning three decades. We find that fetal bowel abnormalities in MVID occurred in up to 60% of cases of MVID for which prenatal ultrasonography or pregnancy details were reported. Suspected fetal bowel abnormalities appeared in the third trimester of pregnancy and correlated with postnatal, early-onset diarrhea and case-fatality risk during infancy. Fetal bowel dilation correlated with *MYO5B* loss-of-function variants. In conclusion, MVID has already started during fetal life in a significant number of cases. Genetic testing for MVID-causing gene variants in cases where fetal bowel abnormalities are suspected by ultrasonography may allow for the prenatal diagnosis of MVID in a significant percentage of cases, enabling optimal preparation for neonatal intensive care.

## 1. Introduction

Microvillus inclusion disease (MVID; https://omim.org/entry/251850/ (accessed on 7 April 2022)) is a rare enteropathy caused by mutations in the *MYO5B*, *STX3,* or *STXBP2* gene. MVID clinically presents with unstoppable secretory diarrhea, the inability to sustain enteral feeds, and failure to thrive [1,2]. The first days after birth are the most critical and characterized by unexpected, severe and life-threatening dehydration of a hypernatremic and/or hypovolemic nature, excessive loss of body weight, and severe metabolic disturbances [3]. The large amount of watery stools, which can be over 300 mL/kg/day at bowel rest, can be mistaken for urine. Hypovolemia-related temporary ischemia may cause neurological symptoms, including developmental delay, in MVID [4]. MVID is invariably fatal if left untreated. Only supportive treatment is available, involving life-long maintenance of hydration, electrolytes, and nutrition with total parenteral nutrition [5]. Many MVID patients who were reported to have died did not survive the neonatal period or infancy [6], but other MVID patients have reached childhood or adolescence [7,8,9,10]. Although the reason for this heterogeneity in disease progression among patients is not known, adequate neonatal intensive care of MVID patients is expected to improve chances of survival. An early diagnosis is therefore crucial. The diagnosis of MVID, however, is challenging, costly, and impeded by various, time-consuming laboratory work-ups, endoscopy, and expert biopsy analyses, and can take several weeks to months [3,11].

Other congenital diarrheal diseases, in particular *SLC36A3*-mutation-associated congenital chloride diarrhea (CCD) and *SLC9A3*-mutation-associated congenital sodium diarrhea (CSD) [12,13], are characterized by fetal bowel abnormalities and intra-uterine diarrhea [13,14]. These can be suspected by ultrasonography, showing the presence of dilated bowel loops and/or the presence of excess amniotic fluid in the uterus (polyhydramnios) [15]. Importantly, prenatal diagnosis of CCD when suspected by ultrasonography can be achieved through examination of AF composition and/or amniocentesis when a parents is a carrier of CCD-associated gene mutations. 

The *SLC26A3*- and *SLC9A3*-encoded proteins—the chloride-bicarbonate exchanger and the sodium-hydrogen exchanger protein-3 protein, respectively—are targets of the *MYO5B*-encoded myosin Vb protein and are affected in the intestinal epithelium in MVID [16,17]. Accordingly, some MVID patients show fecal chloride and/or sodium concentrations as high as those in CCD and CSD patients [6]. Furthermore, morphological defects of the fetal intestine have been observed in a mouse model of MVID [18]. This raises the question of whether fetal bowel abnormalities can also be expected in MVID. Anecdotal reports of dilated fetal bowel loops and polyhydramnios in patients diagnosed with MVID have been published in scattered cases reports. However, these were believed to be rare [11] and received little attention. Moreover, polyhydramnios can be caused by more common risk factors, such as diabetes, twin or multiple pregnancy, infections, or Rhesus disease. 

Therefore, we performed a comprehensive retrospective study, involving 117 cases of MVID described between 1978 and 2021, in order to determine the prevalence of suspected prenatal bowel abnormalities in MVID and the relationship of these with non-MVID-specific risk factors and MVID-associated genes, as well as their clinical significance. 

## 2. Materials and Methods

### 2.1. Collection of Case Reports

The MEDLINE database (https://pubmed.ncbi.nlm.nih.gov/) (access on 7 April 2022) was searched using the following search string: ((microvill* inclusion disease) OR (microvill* atrophy)) AND case report) in order to collect all published reports on cases of MVID. All retrieved articles were manually screened, and articles which did not report on cases of MVID, or which were not digitally accessible, were excluded. Data from the remaining cases were manually combed with regard to reporting of the presence or absence of antenatal/prenatal abnormalities by ultrasonography or other complications during pregnancy. To collect cases of *MYO5B*-associated intrahepatic cholestasis, the MEDLINE database was searched using the search string: (MYO5B AND intrahepatic cholestasis). Retrieved articles were manually screened, and non-relevant articles (i.e., those not reporting on cases of isolated intrahepatic cholestasis in patients with *MYO5B* mutations) were omitted. 

### 2.2. Statistical Analysis

(Prenatal) lifetime prevalence was calculated over the period of 1978–2022 by dividing the number of reported MVID cases with suspected prenatal bowel abnormalities by the total number of MVID cases for which the presence or absence of suspected prenatal bowel abnormalities was reported or, alternatively, by the total number of MVID cases reported. Other statistical analyses were essentially performed as described earlier [6]. Odds ratios (OR) were calculated as measures of association. For statistical analyses, we used the chi-squared test with Yates’ correction to prevent the overestimation of statistical significance for small data and Student’s or Welch’s two-tailed t-tests. Statistical significance was assumed when *p* < 0.05. For non-normally distributed data sets, the Mann–Whitney U test was used. To determine the homogeneity of variances, Levene’s test was used. To determine the normality of the data sets, the Shapiro–Wilk test was used. Sex-corrected birthweight percentiles were calculated using https://www.audipog.net/Courbes-morpho/ (accessed on 7 April 2022). Note that, due to the retrospective nature of this study, data were missing in several variables because of incomplete data recording. However, because our aim was to simply describe and compare subgroups of cases of MVID for which prenatal events were or were not reported, no method for missing data was utilized.

## 3. Results

### 3.1. Lifetime Prevalence of Prenatal Bowel Abnormalities Suspected by Ultrasonography in MVID

The literature search retrieved 144 articles (https://tinyurl.com/3thurukw/ (accessed on 7 April 2022)). Of these, articles that did not report cases of MVID, or that were not digitally accessible, were omitted. A total of 78 articles describing 117 patients diagnosed with MVID between 1978 and 2022 (https://tinyurl.com/ywhxrhe8/ (accessed on 7 April 2022)) were included (Figure 1A). The possibility that some patients may have been reported multiple times could not be excluded. From these 117 cases of MVID, we retrieved 72 cases of MVID for which the presence or absence of prenatal symptoms and/or pregnancy complications was stated (Appendix A; Figure 1A). In 47 of these 72 cases, the presence or absence of bowel dilation and/or polyhydramnios was stated (Appendix A; Figure 1A), and in 25 of these 72 cases, the presence or absence of pregnancy complications was stated (Appendix A; Figure 1A). Some cases reported bowel dilation and/or polyhydramnios while reporting uneventful pregnancy for the same case [19,20]. The stated absence of pregnancy complications, as such, is therefore not a reliable indicator for the absence of fetal bowel dilation and/or polyhydramnios. We therefore restricted our subsequent analyses to the 47 cases for which ultrasonography results (i.e., the presence or absence of bowel dilation, echogenic bowel, and/or polyhydramnios) was explicitly stated (Table 1; Appendix A). 

In 28 (60%) of these 47 cases (ranging from 1989–2022), the presence of prenatal bowel abnormalities suspected by ultrasonography was reported (Table 1), without preference for sex (13 males, 15 females) or consanguinity (35% consanguinity, 31% non-consanguinity) (Appendix A). These included polyhydramnios (75%; 21/28 cases), echogenic bowel (7%; 2/28 cases), and/or dilated bowel loops (57%; 16/28 cases) (Figure 1B). Isolated polyhydramnios (i.e., without report of dilated bowel loops) was reported in 36% (10/28) of cases (Figure 1B). In five cases with polyhydramnios, the amniotic fluid index (AFI) or amniotic fluid volume (AFV) was reported. These all classified as mild polyhydramnios (i.e., AFI 25–30 cm, AFV 7–11 cm). The occurrence of isolated dilated bowel loops (i.e., without reported polyhydramnios) was reported in 21% (6/28) of cases (Figure 1B). Dilated bowel loops and polyhydramnios were co-reported in 36% (10/28) of cases (Figure 1B). In all but one of the cases in which the time of ultrasonographic examination was mentioned (*n* = 13), ultrasonographic abnormalities were found in the third trimester of pregnancy, ranging between the gestational ages of 29 and 36 weeks (mean: 34 weeks). For two of these cases, the suspected bowel abnormalities were explicitly reported to be absent in the first two trimesters. 

In order to investigate trend development over the 3 studied decades, (prenatal) lifetime prevalence analyses of the 47 cases were performed for the 3 sequential decades. Prevalence of prenatal bowel abnormalities suspected by ultrasonography in the periods of 1989–1999, 2000–2010, and 2011–2022 was 33% (3/9), 43% (3/7), and 71% (22/31), respectively (Table 1C). The percentage of all cases of MVID in which prenatal ultrasonography results were mentioned was 10% (1/10) between 1989 and 1999, 8% (3/37) between 2000 and 2010, and 70% (22/31) between 2010 and 2022. The sharp increase in the reporting of prenatal ultrasonography results associated with cases of MVID in the last decade may reflect the increasing trend in ultrasound use for low-risk pregnancies [21,22]. In the period 1989–1999, polyhydramnios, echogenic bowel, and dilated bowel loops were reported in 100% (3/3), 0% (0/3), and 0% (0/3) of the cases, respectively. In the period 2000–2010, polyhydramnios, echogenic bowel, and dilated bowel loops were reported in 66% (2/3), 33% (1/3), and 66% (2/3) of the cases, respectively. In the period 2010–2022, polyhydramnios, echogenic bowel, and dilated bowel loops were reported in 77% (17/22), 0.5% (1/22), and 64% (14/22) of the cases, respectively (Table 1C). Together, these results indicate a trend of increasing prevalence of prenatal bowel abnormalities suspected by ultrasonography over the last three decades. 

A reporting bias cannot be excluded. Indeed, for 45/117 cases of MVID, there was no statement about the presence or absence of suspected fetal bowel abnormalities or the pregnancy period. To calculate the minimum prevalence, it was assumed that in all cases where the presence or absence of prenatal bowel abnormalities or pregnancy details were not reported, such abnormalities would have been absent. This resulted in a prevalence of bowel abnormalities suspected by ultrasonography in MVID fetuses of 25% (28/117) over the period 1989–2022, with a trend of increasing prevalence over the last 3 decades (1989–1999: 10%, 2000–2010: 8%, and 2011–2022: 48%; Table 1D).

In the cases of MVID associated with polyhydramnios, known risk factors for polyhydramnios, such as gestational diabetes, twin or multiple pregnancy, infections, or Rhesus disease were not reported, supporting a possible bowel defect as the underlying cause. This is further supported by the observation of concurrent dilated bowel loops in 48% (10/21) of the cases where polyhydramnios was reported (Figure 1). In those cases where details about the ultrasonographic appearance of bowel dilatations were provided, these were described as generalized bowel dilatation—without the typical, bulbous appearance associated with local obstruction. Accordingly, after birth, physical bowel obstructions that would require surgery (e.g., atresia) were not observed in any of the cases of MVID associated with suspected prenatal bowel abnormalities. 

The estimated prevalence of polyhydramnios in the general population is approximately 1:100 (1%) [23] and approximately 1:5000 (<0.02%) for dilated bowel loops [24]. Therefore, MVID appears associated with a higher risk of polyhydramnios and/or non-obstructive fetal bowel dilation.

### 3.2. Correlations between Polyhydramnios and Fetal Bowel Dilation with MVID-Associated Gene Variants

MVID can be caused by bi-allelic variants in the *MYO5B* [25] or *STX3* [26] gene. MVID-associated *MYO5B* and *STX3* variants were first reported in 2008 and 2013, respectively. A total of 6 of the 28 cases of MVID with suspected prenatal bowel abnormalities occurred prior to 2008, and 8 occurred prior to 2013. Since 2008, gene variants have been reported in 57% (16/28) of MVID cases with prenatal bowel abnormalities suspected by ultrasonography. Of these, 14 carried bi-allelic *MYO5B* variants, and 2 carried bi-allelic *STX3* variants. In the cases of MVID associated with *STX3* mutations, only polyhydramnios (and no bowel dilation) have thus far been reported (Appendix A). All variants in the *MYO5B* or *STX3* gene that have been identified in MVID patients presenting with prenatal bowel abnormalities, and those identified in patients in which the absence of prenatal bowel abnormalities was reported, are shown in Figure 2. 

In the cases of MVID for which the presence of prenatal bowel abnormalities was reported, 10 of the 14 (71%) *MYO5B* mutations were nonsense or frameshift mutations predicted to cause premature termination codons and the loss of myosin Vb protein. Bi-allelic nonsense mutations in *MYO5B* were more prominent in cases of MVID for which the presence of fetal bowel abnormalities was reported. Patients with bi-allelic nonsense *MYO5B* mutations, for which the presence of fetal bowel abnormalities was reported, presented with fetal bowel dilations. Patients with bi-allelic missense *MYO5B* mutations, for which the presence of fetal bowel abnormalities was reported, presented with isolated polyhydramnios. For one patient with bi-allelic nonsense mutations in *MYO5B,* the absence of fetal bowel abnormalities was reported. However, in this patient, a large amount of amniotic fluid at birth was reported. 

MVID can present in some patients diagnosed with familial hemophagocytic lymphohistiocytosis (FHL) type 5, which is a rare, inherited, hyperinflammatory syndrome caused by mutations in the *STXBP2* gene [27,28,29]. In 2 of the reported 24 FHL type 5 patients presenting with MVID-associated symptoms (i.e., severe congenital diarrhea), prenatal bowel abnormalities were suspected by ultrasonography (echogenic bowel and dilated bowel). No ultrasonography results were reported for the other patients [27,28,30,31].

Bi-allelic *MYO5B* gene variants, with the exclusion of bi-allelic variants that cause premature termination codons or the loss of the RAB11A-binding site at the carboxyl-terminus of the myosin Vb protein, can also cause isolated intrahepatic cholestasis with no or only mild intestinal symptoms [32,33,34]. For none of the 33 patients with *MYO5B*-associated, isolated, intrahepatic cholestasis (https://tinyurl.com/2p87v6tj (accessed on 7 April 2022)) were prenatal bowel abnormalities suspected by ultrasonography, nor were other complications during pregnancy reported. 

Together, the occurrence of polyhydramnios associated with MVID was not restricted to either the *MYO5B*, *STX3,* or *STXBP2* variants. With regard to *MYO5B*, suspected prenatal bowel abnormalities were restricted to MVID-associated *MYO5B* variants and were, therefore, associated with bowel disease, rather than with pathogenic *MYO5B* variants as such. 

### 3.3. Correlations between Suspected Fetal Bowel Abnormalities and Case-Fatality during Infancy and MVID Morbidity

We next determined whether the suspected presence of bowel abnormalities before birth correlated with case-fatality risk during infancy, the timing of delivery, and birthweight (percentile), and the onset, extent, and composition of diarrhea after birth. 

#### 3.3.1. Case-Fatality Risk during Infancy

Many patients with MVID who were reported to have died did not survive infancy (the period from birth up to 1 year of age), while other patients with MVID survived infancy and reached childhood or adolescence. We determined the correlation between the presence or absence of suspected fetal bowel abnormalities and the case-fatality risk during infancy. The case-fatality risk during infancy was presented by the number of MVID patients who were reported to have died between birth and 1 year of age, divided by the total number of MVID patients for whom survival data for the infancy period was reported (over the period 1989–2022). Cases with suspected fetal bowel abnormalities showed a statistically significant increase in case-fatality risk during infancy when compared to cases without suspected fetal bowel abnormalities (*X*^2^ (1, *N* = 30] = 3.99, *p* = 0.017) (Figure 3A).

#### 3.3.2. Preterm Birth and Birthweight

MVID is associated with high risk of preterm birth and lower birthweight [6,35]. Polyhydramnios has also been associated with a higher risk of preterm birth [36]. The cases of MVID in which prenatal polyhydramnios and/or bowel dilation was observed demonstrated no significant difference in preterm birth (*X*^2^ (1, *N* = 38] = 0.05, *p* = 0.82). MVID-associated prenatal polyhydramnios and/or bowel dilation did not correlate with birthweight (2919 g ± 556 g versus 3119 ± 574 g, respectively; t (43) = −0.97, *p* = 0.17), when compared to the cases of MVID in which neither polyhydramnios nor dilated bowel were observed (Figure 3B). 

#### 3.3.3. Postnatal Diarrhea: Onset, Volume, and Electrolyte Composition

MVID has been divided into 2 subtypes based on the early-onset of diarrhea (within days after birth) or the late-onset of diarrhea (>2 months after birth) [37]. In 84% of the cases of MVID that were associated with bowel dilation and/or polyhydramnios, the onset of diarrhea was within the first week of life. This occurred in 60% of cases of MVID with neither dilated bowel nor polyhydramnios. In 92% and 80% of cases of MVID with or without suspected fetal bowel abnormalities, respectively, the onset of diarrhea was within the first 2 weeks of life. None of the cases of MVID with suspected fetal bowel abnormalities displayed a late-onset of diarrhea (>2 months after birth), whereas 20% of the cases of MVID without suspected fetal bowel abnormalities were associated with late-onset MVID (Figure 3C). Although the data set is relatively small, the current data suggest that the presence of suspected fetal bowel abnormalities correlated with the early-onset presentation of diarrhea.

There was no statistically significant difference in postnatal stool volume between cases of MVID in which prenatal polyhydramnios and/or bowel dilation was observed when compared to cases of MVID in which neither polyhydramnios nor dilated bowel were observed (227 ± 98 mL/kg/day (*n* = 11) versus 176 ± 49 mL/kg/day (*n* = 6), respectively; t (15) = 1.13, *p* = 0.14) (Figure 3D).

Polyhydramnios and/or prenatal bowel dilation is a prominent feature of CCD and CSD, which are characterized by high (>90 mEq/L) fecal chloride or sodium concentrations, respectively. We therefore hypothesized that the presence of polyhydramnios and/or prenatal bowel dilation in MVID patients may have correlated with postnatal fecal electrolyte concentrations. However, there was no statistically significant difference in fecal sodium and chloride concentrations between cases of MVID in which prenatal polyhydramnios and/or bowel dilation was observed when compared to cases of MVID in which neither polyhydramnios nor dilated bowel were observed (Figure 3E,F; for fecal sodium: 99 ± 26 mEq/L (*n* = 13) versus 89 ± 32 mEq/L (*n* = 10), respectively; t (21) = 0.80, *p* = 0.22); for fecal chloride: 84 ±27 mEq/L (*n* = 8) versus 82 ± 26 mEq/L (*n* = 8), respectively; t (14) = 0.11, *p* = 0.46). In support, of the 7 cases of MVID with reported fecal chloride concentrations of >90 mEq/L, 3 presented with prenatal polyhydramnios and/or bowel dilation and two did not (Appendix A). Similarly, of the 13 cases of MVID associated with fecal sodium concentrations of >100 mEq/L, 7 presented with prenatal polyhydramnios and/or bowel dilation and 6 did not (Appendix A).

Taken together, prenatal bowel abnormalities suspected by ultrasonography were restricted to cases of the early-onset form of MVID and did not correlate with gestational age, birthweight, fecal volume, or stool electrolyte composition, but they did correlate with an increased case-fatality risk during infancy.

## 4. Discussion

Anecdotal reports of MVID-associated prenatal bowel abnormalities suspected by ultrasonography have been published, but these abnormalities were considered to be rare [11]. In this retrospective case study spanning 33 years, we now demonstrate that such prenatal bowel abnormalities were not rare, but rather occurred in 60% (28/47) of cases of MVID between 1989 and 2022. On a cautionary note, this number was based on cases of MVID for which the presence or absence of prenatal symptoms and/or pregnancy complications was reported (62% (72/117) of all reported cases), and a reporting bias cannot be excluded. Nonetheless, in a scenario in which in all of the remaining cases for which the presence or absence of prenatal bowel abnormalities or pregnancy complications was not reported, such abnormalities would have been absent and the percentage of MVID fetuses with bowel abnormalities suspected by ultrasonography would have been 24% (28/117) (and when considering fetuses presenting with only bowel dilation, 14% (16/117)). This is significantly higher when compared to the world average percentages of polyhydramnios (<1%) [25] or fetal bowel dilatation (<0.02%) [26]. Moreover, when restricting the analyses to the last decade (2010–2022) the prevalence would be 70% (21/30) when including only cases where the presence or absence of prenatal bowel abnormalities was reported, and at least 46% (21/46) when assuming that, for all remaining cases in that period, prenatal bowel abnormalities or pregnancy complications would have been absent. Thus, MVID fetuses displayed higher odds of presenting with prenatal bowel abnormalities as suspected by ultrasonography. 

The timing of ultrasonographic examination may be a factor in determining the prevalence of prenatal bowel abnormalities. Ultrasonographic examination to monitor fetal health became routine in the late 1970s, at the time when the first cases of MVID were being reported [1]. Ultrasonographic examination is most commonly performed in the first trimester and in the second trimester at 18–20 weeks of gestation. In cases of MVID, ultrasonographic bowel abnormalities were detected in the third semester (median 34 weeks of gestation (range 29–36)). Further, in some cases where abnormalities were detected in the third trimester, it was explicitly noted that these abnormalities had not been observed in the first or second trimester. When taking into account that fetuses start swallowing amniotic fluid and show gastric peristalsis from gestational week 25, the onset of polyhydramnios and/or dilated bowel loops in the third trimester supports an underlying bowel defect. Because ultrasonographic examination in the third trimester is not routine for low-risk pregnancies in many countries, the prevalence of prenatal bowel abnormalities associated with MVID may be underestimated. 

The presence of prenatal bowel abnormalities suspected by ultrasonography in cases of MVID correlated with the postnatal early-onset of diarrhea and an increased case-fatality risk during infancy. The presence of prenatal bowel abnormalities suspected by ultrasonography in cases of MVID did not correlate with preterm birth or birthweight. This is in agreement with the notion that preterm birth and birthweight in MVID appeared to be associated with *MYO5B* mutations, rather than with bowel dysfunctions as such [6], whereas our results suggest that the occurrence of suspected fetal bowel abnormalities in MVID was directly related to bowel disease rather than to variants in specific MVID-associated genes (i.e., *MYO5B, STX3,* or *STXBP2*). This is in agreement with the notion that these three genes cooperate in the same cellular pathway that controls electrolyte exchange at the brush border of intestinal epithelial cells [29,38]. The relatively milder variations in *MYO5B* that cause intrahepatic cholestasis with no or only mild intestinal symptoms [29,34,39] were not associated with fetal bowel abnormalities, which underscores the association of the latter with a certain level of bowel dysfunction. Interestingly, two other congenital diarrheal enteropathies, congenital tufting enteropathy (CTE; https://tinyurl.com/4j9ue6yx (accessed on 7 April 2022))) and trichohepatoenteric syndrome (THES; https://tinyurl.com/39429uty (accessed on 7 April 2022)) were not associated with prenatal bowel abnormalities suspected by ultrasonography. The observation that the intestinal chloride–bicarbonate exchanger and the sodium–hydrogen exchanger protein-3 protein—which are mutated in CCD and CSD, respectively—are regulated by *MYO5B* (but not CTE- or THES-associated genes) and were mislocalized in the postnatal MVID intestine [16,17] suggests that the fetal bowel abnormalities in MVID and in CCD/CSD may share the same etiology. Although such detrimental effects of *MYO5B* mutations on the expression and/or localization of the chloride–bicarbonate exchanger and sodium–hydrogen exchanger protein-3 protein in the fetal intestine needs to be confirmed, it can be expected that the inability to absorb chloride and/or sodium from the fetal intestinal lumen perturbs the osmotic gradient between lumen and mucosa, and, consequently, the lumen-to-mucosa flux of the hypotonic amniotic fluid. It must be noted that not all cases of MID with postnatal fecal electrolytes above 90 or 100 mEq/L (characteristic of CCD and CSD, respectively) displayed polyhydramnios or fetal bowel dilation, which suggests that other—yet to be defined—mechanisms may also play a role. How the growing fetus maintains body-water homeostasis, when amniotic fluid absorption in the intestine is impaired, is not known. 

In conclusion, here we find that MVID had already started during fetal life in a significant number of cases. We suggest considering MVID—in addition to CCD and CSD—as a potential cause of non-obstructive fetal bowel abnormalities suspected by ultrasonography. The trend of the increasing prevalence of fetal bowel abnormalities suspected by ultrasonography over the last three successive decades suggests that this prevalence may hold for the coming years. If so, genetic testing for MVID-causing gene variants in cases where non-obstructive bowel abnormalities are suspected by ultrasonography, and where no other risk factors (e.g., diabetes, twin or multiple pregnancy, infections, or Rhesus disease) are present, may allow for the prenatal diagnosis of MVID between 25% and 60% of all cases (or between 46% and 70% when taking only the last decade into account). This will enable the optimal preparation for neonatal intensive care, the possibility of giving birth in a medical center with an expert pediatric gastroenterology unit, hence contributing to improved postnatal prognoses. 

## Figures and Tables

**Figure 1 jcm-11-04331-f001:**
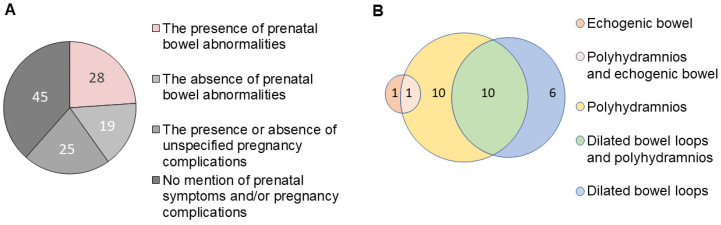
(**A**) Diagram showing the number of cases of MVID between 1989 and 2022 for which the presence or absence of prenatal bowel abnormalities was reported, for which the presence or absence of unspecified pregnancy complications was mentioned, and for which there was no mention of prenatal symptoms and/or pregnancy complications; (**B**) Diagram showing the number of cases of MVID between 1989 and 2022 for which the presence of prenatal bowel abnormalities were reported, specified to the presence of echogenic bowel, polyhydramnios, and/or fetal bowel dilation.

**Figure 2 jcm-11-04331-f002:**
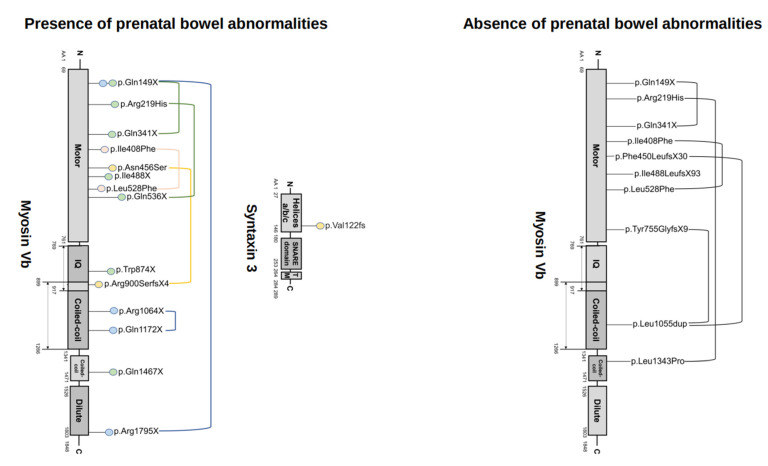
Diagram showing the variants in MYO5B and STX3 that have been identified in cases of MVID in which the presence (**left side**) or absence (**right side**) of prenatal bowel abnormalities suspected by ultrasonography were reported. The color-coded circles indicate the types of fetal bowel abnormalities (using the same color code as in Figure 1). Lines drawn between gene variants indicate that these variants occurred in the same individual.

**Figure 3 jcm-11-04331-f003:**
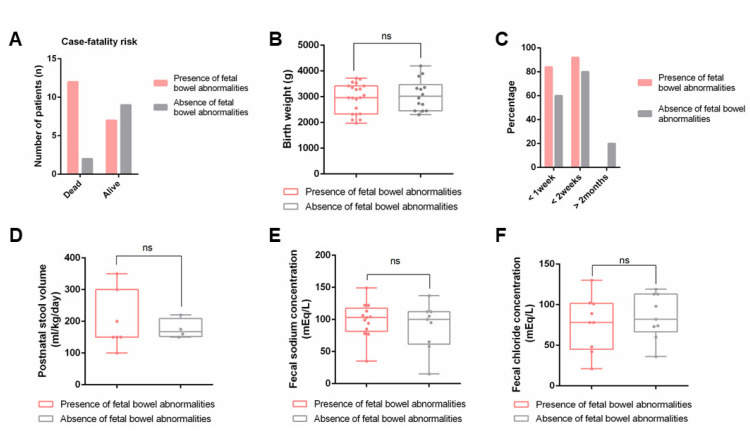
(**A**) Results of the chi-squared test with Yates’ correction to determine the difference in case-fatality risk during infancy as a function of the presence or absence of suspected fetal bowel abnormalities. (The Fisher’s exact statistic value was 0.0238); (**B**) Birthweights associated with cases of MVID in which the presence (red) or absence (black) of suspected fetal bowel abnormalities were reported; (**C**) percentage of cases of MVID in which the presence (red) or absence (black) of suspected fetal bowel abnormalities were reported in which diarrhea started within 1 week, within 2 weeks, or after 2 months; (**D**–**F**) Postnatal stool volumes (**D**), fecal sodium (**E**), and chloride (**F**) concentrations with cases of MVID in which the presence (red) or absence (black) of suspected fetal bowel abnormalities were reported.

**Table 1 jcm-11-04331-t001:** Cases of MVID with reported presence or absence of prenatal bowel abnormalities suspected by ultrasonography (number, prevalence, and as a percentage of all cases reported in the indicated period).

	A	B	C	D
Time period	Cases of MVID with reported presence of suspected prenatal bowel abnormalities (*n*)	Cases of MVID with reported absence of suspected prenatal bowel abnormalities (*n*)	Prevalence, when *n* (column A) is divided by the total number of MVID cases for which the presence or absence of suspected prenatal bowel abnormalities was reported in the specified time period (%)	Prevalence, when *n* (column A) is divided by the total number of MVID cases in the specified time period (%)
1989–2022	28	19	60% (28/47)	25% (28/114)
1989–1999	3	6	33% (3/9)	10% (3/30)
2000–2010	3	4	43% (3/7)	8% (3/38)
2011–2022	22	9	71% (22/31)	48% (22/46)

## Data Availability

No new data were generated in this study.

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
