# Peer review of "Fetal Bowel Abnormalities Suspected by Ultrasonography in Microvillus Inclusion Disease: Prevalence and Clinical Significance"

_jcm, 2022, doi:10.3390/jcm11154331_

Round 1

Reviewer 1 Report

This is an interesting manuscript, exploring one prenatal symptom to the rare intestinal failure disease MVID. This is a retrospective review of the literature to date. The authors bring to light this important finding for the first time in the literature. 

In the methods section this statement should be further explained, "After manual 

inspection and removal of non-relevant articles". Clear rationale for exclusion of manuscripts should be provided. 

Figure 1 is helpful, a similar size based graphic of those cases where no polyhydramnios was seen, and those where polyhydramnios was not discussed should also be included in Figure 1 for comparison. 

It is important that reporting bias is considered, where in lack of the discussion of presence or absence of polyhydramnios is assumed to be absence in the results section. 

In the introduction and discussion, more common etiologies of polyhydramnios should also be discussed, such as diabetes. This is relevant, screening for genetic defects in patients with polyhydramnios and diabetes is likely not necessary/cost effective. The current version of the manuscript would inadvertently imply this. 

Figure 3 shows mostly negative data. The authors should consider correlation with other disease outcomes such as mortality, % of TPN calories vs enteral calories (enteral autonomy) as markers of disease severity. Authors should also provide discussion of the negative data in the discussion. 

Author Response

Point by point reply to the reviewers

Reviewer 1.

  1. In the methods section this statement should be further explained, "After manual 

inspection and removal of non-relevant articles". Clear rationale for exclusion of manuscripts should be provided. 

Our reply: We have clarified the manuscript exclusion criteria in the Methods section as follows: “All retrieved articles were manually screened and articles that did not report on cases of MVID or were not digitally accessible were excluded.”  (lines 87-89 in the revised version with track changes)

  1. Figure 1 is helpful, a similar size based graphic of those cases where no polyhydramnios was seen, and those where polyhydramnios was not discussed should also be included in Figure 1 for comparison. 

Our reply: We have included a new figure depicting the requested information. This is now Figure 1A. Figure 1 in the original manuscript is now Figure 1B.

  1. It is important that reporting bias is considered, where in lack of the discussion of presence or absence of polyhydramnios is assumed to be absence in the results section. 

Our reply: We have now included in the Results section the minimum prevalence data when corrected for reporting bias, i.e., assuming that in all cases without statements on suspected fetal bowel abnormalities such abnormalities would have been absent (lines 176-183 in the revised version with track changes).   

  1. In the introduction and discussion, more common etiologies of polyhydramnios should also be discussed, such as diabetes. This is relevant, screening for genetic defects in patients with polyhydramnios and diabetes is likely not necessary/cost effective. The current version of the manuscript would inadvertently imply this. 

Our reply:  We agree with the reviewer that this should be better highlighted and have discussed the more common etiologies of polyhydramnios in the Introduction section and Discussion section of the revised manuscript (lines 72-75,  80, 396-398 in the revised version with track changes).

  1. Figure 3 shows mostly negative data. The authors should consider correlation with other disease outcomes such as mortality, % of TPN calories vs enteral calories (enteral autonomy) as markers of disease severity. Authors should also provide discussion of the negative data in the discussion. 

Our reply:  We thank the reviewer for the suggestion. We have determined the correlation between the presence or absence of suspected fetal bowel abnormalities and case-fatality risk during infancy. The infancy period is defined as the period from birth up to 1 year of age. Infant case-fatality risk is presented by the number of MVID patients who have died between birth and 1 year of age, divided by the total number of MVID patients for who survival data for the infancy period was available (in the period 1989-2022). Cases with suspected fetal bowel abnormalities showed a statistically significant increase in case-fatality risk during infancy, when compared to cases without suspected fetal bowel abnormalities (65% vs 23%, respectively; p<0.05). We have included these new results in the Results and Discussion sections (lines 249, 251-261, 318-319, 357-359 in the revised version with track changes) and in new Figure 3A. For the other suggested marker of disease severity, i.e., % of TPN calories vs enteral calories, there was unfortunately not enough data available to determine correlations. Most patients are on 100%  TPN.  Discussion of the negative data has been included in the revised manuscript (lines 357-366 (preterm birth and birth weight), lines 385-389 (fecal electrolyte values) in the revised version with track changes)

Reviewer 2 Report

The article entitles "Fetal Bowel Abnormalities Suspected by Ultrasonography in Microvillus Inclusion Disease: Prevalence and Clinical Significance" is review of literature about Fetal Bowel Abnormalities in MVID patient. It complete previous articles like preterm birth in MVID patient and show the interest of reevaluation of phenotype in rare disease. The article is clear, well written and illustrated. The only (minors) remarks is that the introduction is a bit too long and some part notably about CSD and CCD could be shortened, moreover in discussion some comparison of Fetal bowel abnormalities suspected by ultrasonography with other severe diarrhea  like tufting enteropathy or tricho-hepato-enteric syndrom could be useful.

Author Response

Reviewer 2

  1. The only (minors) remarks is that the introduction is a bit too long and some part notably about CSD and CCD could be shortened

Our reply: We have shortened this part per the reviewer’s suggestion. We would like to leave a part of CCD/CSD in as it was the rationale for studying the prevalence of prenatal bowel abnormalities suspected by ultrasonography in MVID (lines 57-60 in the revised version with track changes).

  1. In discussion some comparison of Fetal bowel abnormalities suspected by ultrasonography with other severe diarrhea  like tufting enteropathy or tricho-hepato-enteric syndrome could be useful.

Our reply: We have included a comparison with congenital tufting enteropathy (CTE) and tricho-hepato-enteric syndrome (THES). For congenital tufting enteropathy, we collected all case reports in PubMed using the search string (congenital tufting enteropathy AND case report). From the resulting articles (tinyurl.com/4j9ue6yx), we retrieved 49 cases. For 42 of these cases (86%) there was no mention of prenatal findings or pregnancy details. For 7 cases there was mention of the presence or absence of prenatal ultrasonography and/or presence or absence of pregnancy complications. Of 1 of these there was mention of the presence of polyhydramnios. The low percentage of CTE case reports mentioning the presence or absence of prenatal bowel abnormalities suspected by ultrasonography makes it difficult to draw conclusions. For tricho-hepato-enteric syndrome (THES), we collected all case reports in PubMed using the search string (tricho-hepato-enteric syndrome AND case report) From the resulting articles (tinyurl.com/39429uty), we retrieved 35 cases. For 33 of these cases (94%) there was no mention of prenatal ultrasonography findings or pregnancy details, except for 15 cases in which intra-uterine growth restriction (IUGR) was reported. Suspicion of IUGR is typically confirmed by ultrasonography. For none of the cases of THES there was mention of polyhydramnios and/or fetal bowel dilation suspected by ultrasonography. THES therefore does not appear to be associated with fetal bowel abnormalities as suspected by ultrasonography. Given that MVID-associated myosin Vb but not the CTE- and THES-associated gene products EPCAM, SPINT, TTC37 and SKIVL2 have been functionally linked to the CCD and CSD-associated gene products SLC26A3 and SLC9A3, these comparisons of distinct congenital diarrheal enteropathies suggest that fetal bowel abnormalities seen in MVID may be linked to SLC26A3 and/or SLC9A3. We have included a shortened version of the above in the revised manuscript (lines 371-375 and 377 in the revised version with track changes).